# Proximity biotinylation reveals novel secreted dense granule proteins of *Toxoplasma gondii* bradyzoites

**Santhosh Mukund Nadipuram**[1,2], **Amara Cervantes Thind**[1,3], **Shima Rayatpisheh**[4], **James Akira Wohlschlegel**[3,4], **Peter John Bradley**[1,3]*

**1** Department of Microbiology, Immunology and Molecular Genetics, University of California, Los Angeles, California, United States of America, **2** Division of Pediatric Infectious Diseases, Department of Pediatrics, Cedar-Sinai Medical Center, Los Angeles, California, United States of America, **3** Molecular Biology Institute, University of California, Los Angeles, California, United States of America, **4** Department of Biological Chemistry and Institute of Genomics and Proteomics, University of California, Los Angeles, California, United States of America

* pbradley@ucla.edu

**Data Availability Statement:** All relevant data are within the manuscript and its Supporting Information files.

## Abstract

*Toxoplasma gondii* is an obligate intracellular parasite which is capable of establishing life-long chronic infection in any mammalian host. During the intracellular life cycle, the parasite secretes an array of proteins into the parasitophorous vacuole (PV) where it resides. Specialized organelles called the dense granules secrete GRA proteins that are known to participate in nutrient acquisition, immune evasion, and host cell-cycle manipulation. Although many GRAs have been discovered which are expressed during the acute infection mediated by tachyzoites, little is known about those that participate in the chronic infection mediated by the bradyzoite form of the parasite. In this study, we sought to uncover novel bradyzoite-upregulated GRA proteins using proximity biotinylation, which we previously used to examine the secreted proteome of the tachyzoites. Using a fusion of the bradyzoite upregulated protein MAG1 to BirA* as bait and a strain with improved switch efficiency, we identified a number of novel GRA proteins which are expressed in bradyzoites. After using the CRISPR/Cas9 system to characterize these proteins by gene knockout, we focused on one of these GRAs (GRA55) and found it was important for the establishment or maintenance of cysts in the mouse brain. These findings highlight new components of the GRA proteome of the tissue-cyst life stage of *T. gondii* and identify potential targets that are important for maintenance of parasite persistence *in vivo*.

## Introduction

*Toxoplasma gondii* is an apicomplexan parasite that chronically infects nearly every animal and approximately one-third of the world's human population [1–3]. While the infection is typically asymptomatic in healthy persons, infection in immunocompromised patients (such as those with AIDS or patients taking immunosuppressive drugs) can result in life-threatening

**Funding:** This work was supported by NIH grants AI064616 and AI125106 to P.J.B., and no. GM089778 to J.A.W., the Ruth L. Kirschstein National Research Service Award (T32) AI007323 to S.M.N and A.T. The funders had no role in study design, data collection and analysis, decision to publish, or preparation of the manuscript.

**Competing interests:** The authors have declared that no competing interests exist.

central nervous system disease [3,4]. While therapies exist that can combat the acute infection consisting of rapidly growing tachyzoites, there are no effective treatments that can clear the chronic infection which is mediated by slow-growing bradyzoite cysts. Thus, patients who are chronically infected with bradyzoites live under a life-long threat of reactivation of the parasite if a lapse in immune surveillance occurs [4]. A mechanistic understanding of how *T. gondii* bradyzoite cysts are formed and able to maintain lifelong persistence in the host is critical for the development of novel therapies that target this important intracellular pathogen.

*T. gondii* actively invades its host cells and replicates inside of a membrane-bound parasitophorous vacuole (PV) within the host cell cytoplasm [5]. Host cell invasion, PV formation and maintenance are mediated by a set of specialized secretory organelles known as micronemes, rhoptries, and dense granules [6–9]. While micronemes and rhoptries play roles in the initial stages of attachment and invasion, the dense granules secrete proteins called GRAs into the vacuolar space that participate in the remodeling and maintenance of the PV during intracellular replication [10–16]. While many GRAs function within the vacuole after secretion, some GRAs are able to cross the vacuolar membrane into the host cell and hijack cellular immune functions [17–22]. Most of the currently known GRAs have been characterized in the context of the acutely infectious tachyzoite life-cycle stage of the parasite. As expected, some of these have been found to have important roles during both stages of infection. GRA17, which operates as a small-molecule transporter at the PVM in tachyzoites, has also been shown to be important for growth and maintenance of bradyzoites [23,24]. Several other well-studied GRAs such as GRA1, GRA2, GRA4, GRA5, GRA6, GRA9 and GRA12 have been shown to express during the bradyzoite phase and localize to the cyst wall and matrix at different points of maturation [16,25–28]. Deletion of GRA2 results in defective formation of the cyst matrix and the failure of GRA4 and GRA6 to accumulate at the cyst periphery; this finding was in line with previous data which revealed the importance of GRA2, GRA4 and GRA6 in the formation of the intravacuolar network in tachyzoites [12,16,29].

While cyst formation and maintenance are likely to involve an array of bradyzoite-specific GRA proteins, only a few such proteins have been identified and little is known how they might enable the parasite to establish and maintain the chronic infection. These include CST1, bradyzoite pseudokinase 1 (BPK1) and the newly discovered CST2/GRA50, CST3/GRA51, CST4, CST5/GRA52, and CST6/GRA53 [30–34]. The deletion of the glycoprotein CST1 results in fragile cysts with thin cyst walls and lower cyst burden in a chronic-infection model [31]. BPK1 was found to be important in oral infectivity, and the disruption of this protein led to small *in vivo* cysts whose walls were more easily broken down by pepsin-acid treatment [34]. The purification of the cyst wall using Percoll gradient followed by immunoprecipitation with an anti-CST1 monoclonal antibody was used to identify cyst components. This study revealed 5 new bradyzoite-upregulated GRAs which localized to the tissue cyst wall and cyst matrix (named CST2 –CST6). Of these, CST2/GRA50 was important for parasite virulence and establishment of cyst burden [33]. Although this study revealed new proteins which are bradyzoite-specific cyst-wall components, it is likely that there are many more GRAs which participate in the formation and maintenance of bradyzoite tissue cysts during the chronic *T. gondii* infection.

To further bridge this gap in our knowledge, we have developed new approaches to rapidly identify and determine the function of novel GRAs. One important advance is our adaptation of the BioID technique to *T. gondii* for identifying proteins in specific subcellular compartments [35,36]. This approach utilizes a bait protein fused to a promiscuous biotin ligase (BirA*) that biotinylates interacting and proximal proteins in subcellular compartments *in vivo*. The resulting biotinylated proteins are isolated by streptavidin chromatography and are then identified by mass spectrometry. We have recently utilized this approach to identify a robust dataset of previously known and candidate vacuolar proteins in the tachyzoite stage of

the parasite [36]. Our follow-up endogenous tagging of a subset of these candidate GRAs resulted in the verification of many new GRAs, demonstrating the effectiveness of this approach. This dataset thus provided an initial tachyzoite GRA proteome of *T. gondii*, many of which are also expressed in bradyzoites.

In this manuscript, we build on these results by extending the GRA BioID experiments to bradyzoites, utilizing a bradyzoite-upregulated protein as a bait and a background strain which enables improved switching to bradyzoites *in vitro*. Using this approach, we have identified five novel GRA proteins expressed in bradyzoite cysts. We subsequently used CRISPR/Cas9 to disrupt all five of the genes encoded these GRAs, demonstrating that they are not essential for growth *in vitro*. We then focused on one of these knockouts (GRA55) which is dispensable for growth *in vitro* and virulence of *T. gondii* during the acute phase of infection but is important for establishing or maintaining chronic infection in a mouse model. Together, this research identifies a group of novel bradyzoite-upregulated GRA proteins that are novel potential targets for the chronic *T. gondii* infection.

## Results

### Adaptation of BioID for Toxoplasma bradyzoite GRAs

Our previous success with using BioID to identify secreted GRAs in tachyzoites suggested that this approach would also work for bradyzoites. To improve our changes of obtaining bradyzoite GRAs, we made several modifications to our approach. First, we used as our background the Prugniaud$\Delta$*ku80* strain of *T. gondii* that contains GFP driven by the bradyzoite specific promoter *LDH2*, which has previously been shown to be effective at monitoring switching to bradyzoites [37]. Second, we sought to improve the efficiency of *in vitro* switching to bradyzoites, thus we deleted *AP2IV-4* (S1 Fig), which encodes one of the nuclear Apetala-2 (AP2) proteins which has previously been shown to be important for the maintenance of the tachyzoite phenotype *in vitro* [38]. Lastly, we used the bradyzoite-upregulated GRA protein MAG1 [39,40] as our bait which we fused to the biotin ligase BirA* (Fig 1A). Deletion of *AP2IV-4* and obtaining the Pru$\Delta$*ku80*::$\Delta$*ap2iv-4* and MAG1-BirA*::$\Delta$*ap2iv-4* parasites enables increased bradyzoite switching efficiency without disrupting the lytic cycle of the tachyzoites as previously described [38].

When we expressed the MAG1-BirA* fusion driven from its endogenous promoter, the fusion was readily detectable in tachyzoites and trafficked appropriately to the PV when examined by immunofluorescence assay (IFA) (Fig 1B). While MAG1 is considered a bradyzoite GRA, this detection was expected as it is clearly expressed in tachyzoites and is one of the stronger hits in our tachyzoite GRA BioID experiments [36]. Upon *in vitro* switching to bradyzoites, the fusion expressed in bradyzoites and trafficked appropriately to cyst vacuole where it co-localized with the known GRA marker GRA14 (Fig 1B) [41]. The fusion protein migrated at a molecular weight of ~105 kDa as predicted [42]

### *In vivo* biotinylation of the PV using MAG1-BirA* parasites

After producing switch-efficient MAG1-BirA* parasites, we assessed whether the fusion protein was active in the PV. To do this, we switched MAG1-BirA* parasites *in vitro* using high pH media for 5 days and supplemented biotin to the media on the final day [43]. Transition from the tachyzoite to bradyzoite stage was confirmed by the presence of GFP in the parasite cytosol (Fig 1C). As a control, MAG1-BirA* parasites were switched on a separate monolayer, without biotin supplementation. While the control parasites just showed the expected apicoplast background staining, we observed streptavidin staining overlapping with the fusion protein in the parasite vacuole when we supplemented biotin (Fig 1C). This demonstrated that the

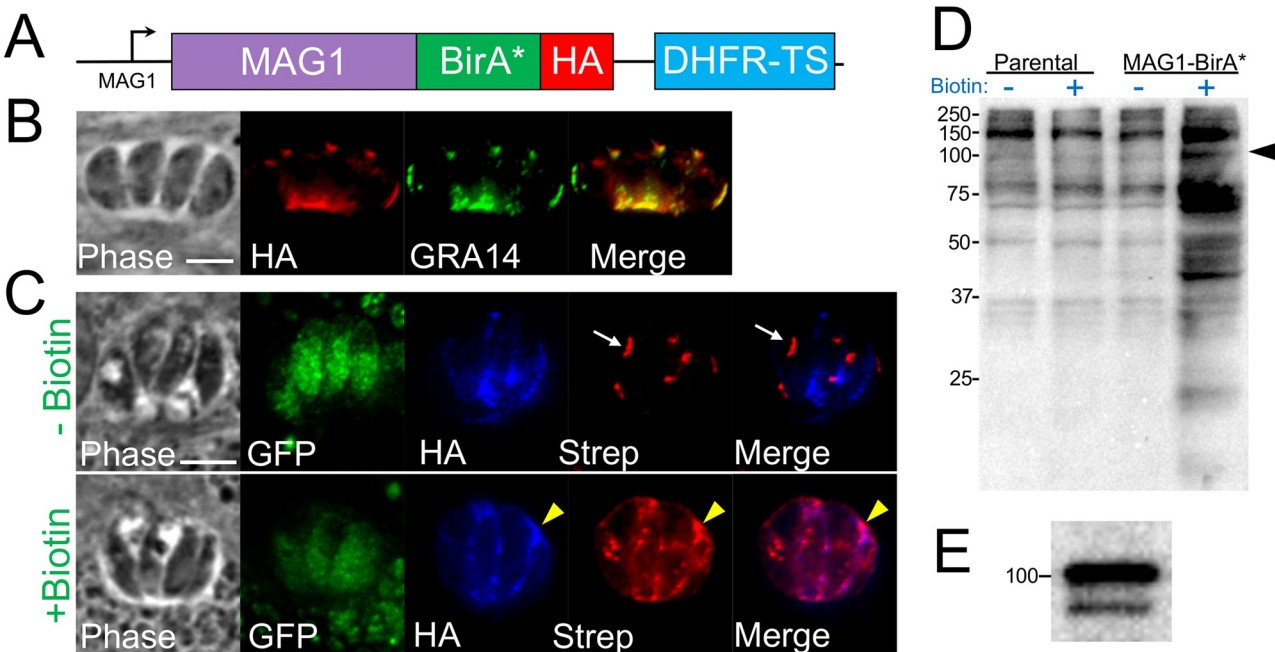

**Fig 1. MAG1-BirA\* localizes to the PV and biotinylates proteins in the cyst vacuole.** (A) Diagram of the construct encoding the promoter and full genomic sequence of MAG1 fused to BirA\* along with a 3xHA C-terminal epitope tag. (B) IFA of MAG1-BirA\* showing that the fusion protein appropriately traffics to the tachyzoite PV and colocalizes with the known protein GRA14. Scale bar: 5 μm (applicable to all panels). (C) IFA of MAG1-BirA\*-expressing parasites, showing the bradyzoite PV is labeled in a biotin-dependent manner (yellow arrowheads, +Biotin row). Endogenously biotinylated apicoplasts are observed with and without biotin (white arrows, -Biotin row). Scale bar: 5 μm (applicable to all panels in C). (D) Western blot of whole-cell lysates of parental (Prugniaud*Δku80Δhxgprt*) and MAG1-BirA\*-expressing parasites -/+ biotin. Lysates were probed with streptavidin-HRP, revealing an increase in biotinylated proteins in MAG1-BirA\*-expressing parasites upon addition of biotin. The MAG1-BirA\* fusion protein is predicted to be ~105 kDa (arrowhead). (E) Western blot showing that MAG1-BirA\* fusion migrates to ~105 kDa as detected by antibody against HA epitope tag.

biotin was able to penetrate the PV compartment and the MAG1-BirA\* was active during the induced bradyzoite stage. We additionally demonstrated by western blot that there was an increase in the number of biotinylated proteins in MAG1-BirA\* lysates when excess biotin was supplemented to the media (Fig 1D). We then proceeded with a large-scale MAG1-BioID experiment by infecting human foreskin fibroblasts with a high multiplicity of infection, switching them *in vitro* to bradyzoites over the course of 5 days, supplementing biotin on day 4. We then harvested the cells, lysed the samples using stringent conditions, and purified the biotinylated protein fraction using streptavidin-affinity chromatography.

As with previous experiments, mass spectrometry analysis revealed a large number of tachyzoite and bradyzoite GRAs highly ranked by peptide spectrum matches when we examined the MAG1-BirA\* lysate in comparison to the parental lysate (Table 1 and S1 Table) [36]. We found the bait protein (MAG1) as well as almost all numbered GRAs, MYR proteins, TgIST1, WNG1, WNG2, bradyzoite GRAs (such as CST proteins, MCP3, MCP4, and BPK1), cyclophilin, NTPaseI and NTPaseII. BPK1 did appear, although slightly lower ranking in abundance. There were several expected proteins that were missing from our dataset including the exported proteins GRA19, GRA20 and GRA21 [44]. In total, the MAG1-BioID experiment revealed 537 proteins that were not present in the control lysate, 675 proteins in total including some GRAs that were present in comparatively low levels in the control lysate (S1 Table). GRAs constituted many of the highest-ranking proteins in this experiment, however, as

**Table 1. Summary of known and novel GRAs found by MAG1-BioID.**

| ToxoDB number | Protein Identification |
|---|---|
| TGME49_209755 | MAG2 |
| TGME49_264660 | SRS44/CST1 |
| TGME49_270240 | MAG1 |
| TGME49_217680 | **GRA57** |
| TGME49_208730 | MCP4 |
| TGME49_204340 | GRA54 |
| TGME49_288650 | GRA12 |
| TGME49_254470 | MYR1 |
| TGME49_212300 | GRA32 |
| TGME49_228170 | IMC2a - GRA44 |
| TGME49_203310 | GRA7 |
| TGME49_289380 | GRA39 |
| TGME49_203290 | GRA34 |
| TGME49_231960 | GRA28 |
| TGME49_232600 | PLP1 |
| TGME49_240060 | TgIST1 |
| TGME49_222370 | SRS13 |
| TGME49_289540 | GRA29 homologue |
| TGME49_208740 | MCP3 |
| TGME49_269690 | GRA29 |
| TGME49_312420 | GRA38 |
| TGME49_230705 | CST3/GRA51 |
| TGME49_239740 | GRA14 |
| TGME49_251540 | GRA9 |
| TGME49_220240 | GRA31 |
| TGME49_226380 | GRA35 |
| TGME49_213067 | GRA36 |
| TGME49_253330 | BPK1 |
| TGME49_208830 | GRA16 |
| TGME49_261650 | CST4 |
| TGME49_232000 | GRA30 |
| TGME49_319340 | CST5/GRA52 |
| TGME49_227620 | GRA2 |
| TGME49_247440 | GRA33 |
| TGME49_309930 | **GRA56** |
| TGME49_308093 | ROP32 |
| TGME49_310780 | GRA4 |
| TGME49_221210 | cyclophilin |
| TGME49_275440 | GRA6 |
| TGME49_290700 | GRA25 |
| TGME49_309760 | **GRA55** |
| TGME49_215220 | GRA22 |
| TGME49_275470 | GRA15 |
| TGME49_304740 | ROP35/WNG1 |
| TGME49_227280 | GRA3 |
| TGME49_260520 | CST6/GRA53 |
| TGME49_203600 | CST2/GRA50 |

*(Continued)*

**Table 1.** (Continued)

| ToxoDB number | Protein Identification |
|---|---|
| TGME49_254720 | GRA8 |
| TGME49_277270 | NTPAse II |
| TGME49_201130 | ROP33 |
| TGME49_270250 | GRA1 |
| TGME49_237015 | GRA43 |
| TGME49_240090 | ROP34/WNG2 |
| TGME49_267740 | GRA48 |
| TGME49_277240 | NTPase I |
| TGME49_230180 | GRA24/TgBRADIN |
| TGME49_254000 | GRA47 |
| TGME49_279100 | MAF1a |
| TGME49_297880 | GRA23 |
| TGME49_237230 | MYR3 |
| TGME49_239010 | TEEGR/HCE1 |
| TGME49_219810 | GRA40 |
| TGME49_236890 | GRA37 |
| TGME49_268790 | **GRA58** |
| TGME49_262050 | ROP39 |
| TGME49_316250 | GRA45 |
| TGME49_234950 | ROP48 |
| TGME49_237880 | GRA13 |
| TGME49_244530 | GRA49 |
| TGME49_286450 | GRA5 |
| TGME49_222170 | GRA17 |

Novel GRAs highlighted in bold and with grey shading.

expected many other secreted proteins including rhoptry proteins (ROPs/RONs), and micro-neme proteins (MICs) were also found concentrated in the top hits of the dataset.

## Identification of novel bradyzoite dense granule proteins from BioID datasets

Our dataset also contained a large number of hypothetical proteins. We narrowed this list of proteins as likely GRAs by selecting those proteins that were highly ranked in the dataset and lacked a C-terminal endoplasmic reticulum retention signal (K/HDEL) [11,45]. We prioritized those proteins with predicted signal peptides, however as many secreted proteins do not contain these sequences, we considered GRA candidates from the hypothetical proteins that also lacked this feature. Because we wished to focus on those proteins that were upregulated in bradyzoites, we also narrowed our candidate pool by focusing on proteins that were shown to be highly transcribed in bradyzoites. However, we also chose some candidates whose expression was similar between tachyzoites and bradyzoites [46]. We chose 9 candidates and used endogenous gene tagging to recombine a C-terminal 3xHA epitope tag for each gene product and examined their localization by IFA. We found that 5 of these trafficked to the PV in both tachyzoites and bradyzoites (Fig 2). We named these GRA55 (TgME49_309760), GRA56 (TgME49_309930), GRA57 (TgME49_217680), and GRA58 (TgME49_268790). In addition, while reviewing earlier datasets from a prior GRA-BioID experiments [36], we found another

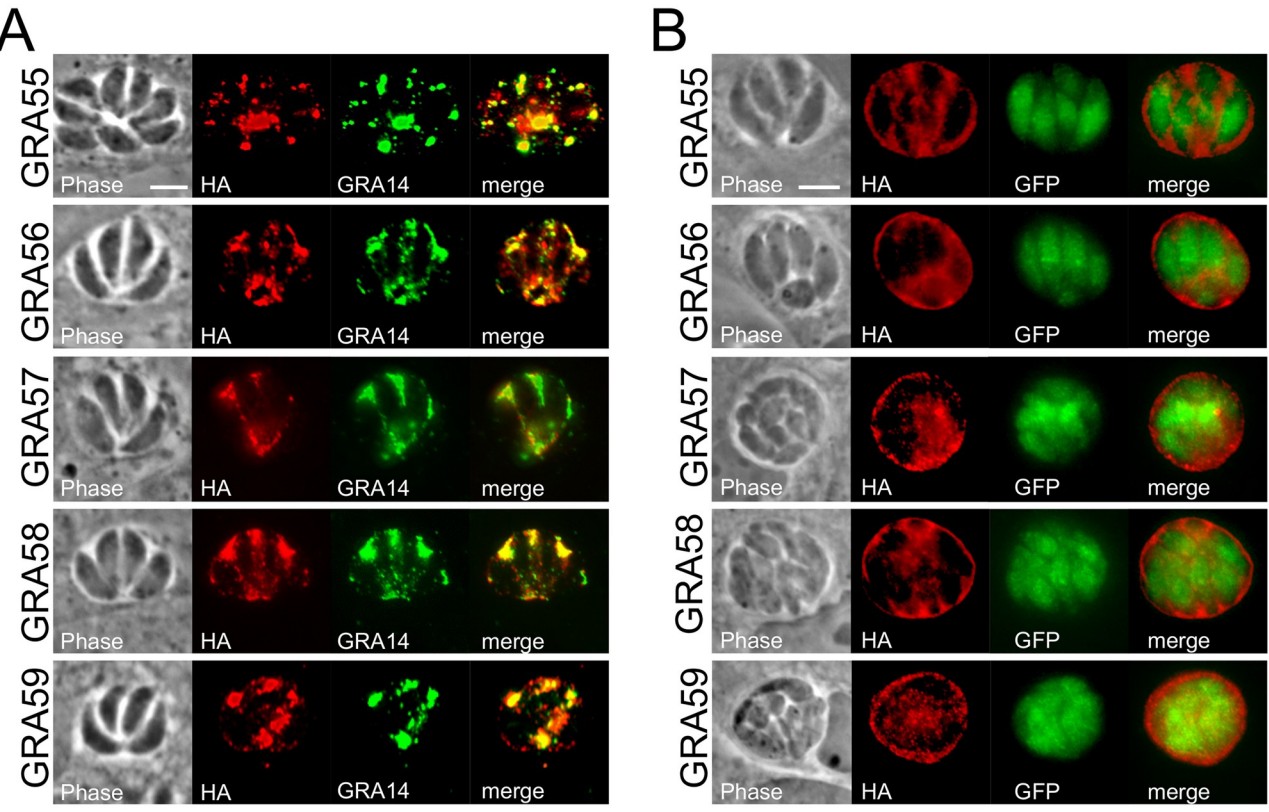

**Fig 2. Identification of novel dense granule proteins, GRA55-GRA59 by MAG1-BioID.** (A) IFA of tachyzoites with rabbit anti-HA antibodies shows strong staining of the parasitophorous vacuole for each novel GRA that colocalizes with GRA14, demonstrating that these are novel dense granule proteins. Gene numbers and novel designations are as follows: GRA55 (TgME49_309760), GRA56 (TgME49_309930), GRA57 (TgME49_217680), GRA58 (TgME49_268790), and GRA59 (TgME49_313440). Scale bar: 5 μm. (B) IFA of bradyzoites with mouse anti-HA antibodies shows staining of each GRA at the cyst periphery and vacuolar space. Transgenic parasites expressed GFP, driven by *LDH*2 promoter, upon successful bradyzoite conversion. Images were taken three days after growth under alkaline-stressed conditions. Scale bar: 5 μm.

GRA–GRA59 (TgME49_313440)–which was also upregulated in bradyzoites. Of these confirmed GRA proteins, only GRA56 had an identifiable domain which has homology to melibiase, an alpha-galactosidase that catabolizes the disaccharide melibiose.

## Targeted deletion of novel GRA proteins and functional characterization

Once we identified these novel GRAs, we assessed their function by gene deletion using CRISPR/Cas9 and homologous recombination [47–49]. Gene knockouts were assessed by loss of the epitope tag from the parental line and replacement of the DHFR selectable marker used in tagging for HXGPRT used in the knockout (e.g. by growth in mycophenolic acid and xanthine but failure to grow in pyrimethamine) (Fig 3A). Each of the knockouts was then confirmed by PCR (Fig 3B). We examined the deletion mutants by plaque assay and discovered none of the lines had impaired *in vitro* growth (Fig 4, S2 Fig). We then proceeded with a pilot study of each of the knockouts to identify severe changes in cyst burden. This pilot study suggested that Δ*gra55* parasites produced substantially lower cyst numbers per mouse brain, thus we focused on this strain for complementation and more detailed *in vivo* analyses.

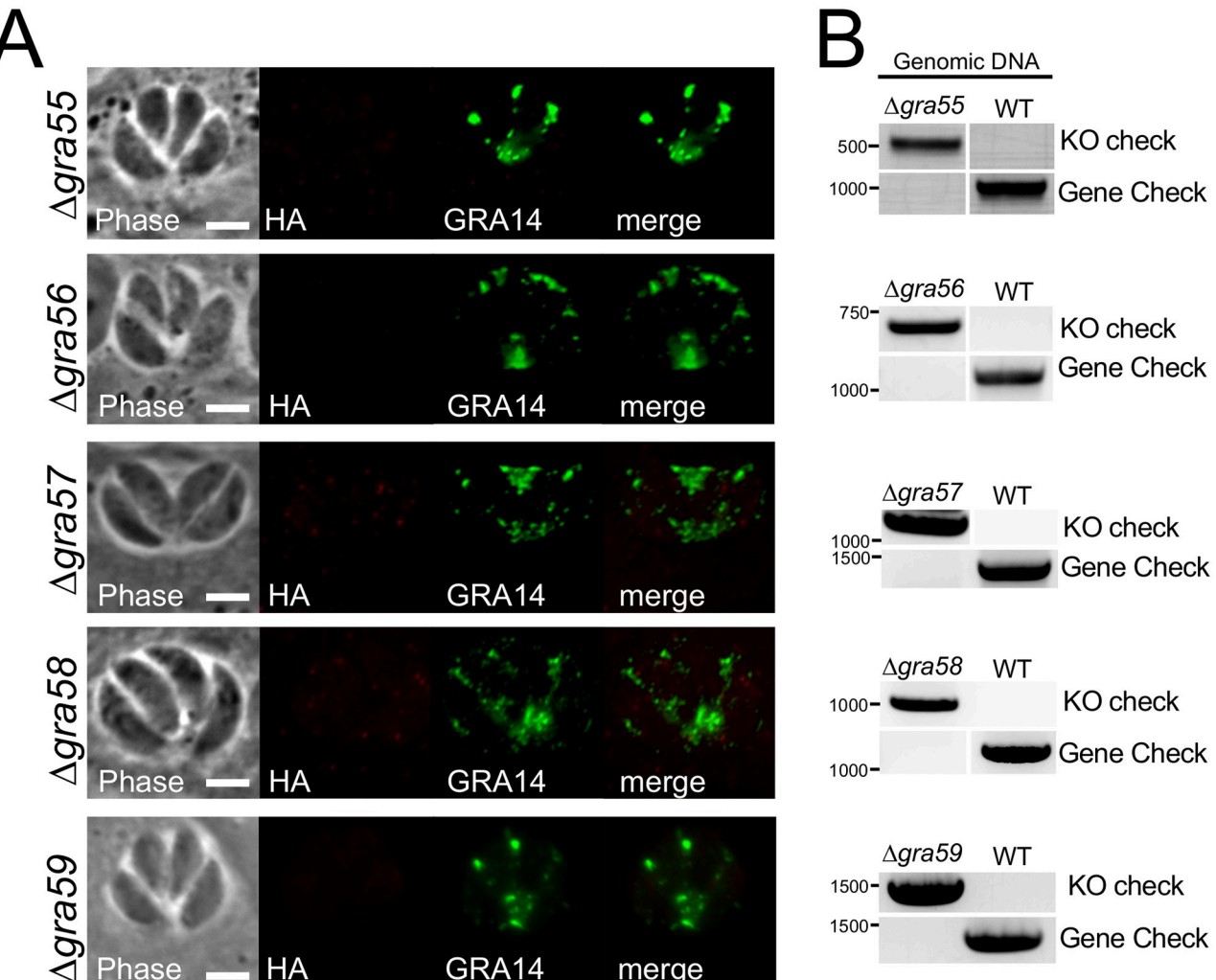

**Fig 3. Deletion of GRAs 55–59 using CRISPR/Cas9.** (A) IFA shows the absence of HA signal in the PV. GRA14 is a marker of the PV and is shown as a control. Scale bar: 5 μm. (B) PCR verification of the deletion of GRAs 55–59. Attempted amplification of genes in knockout parasite strains show an absence of the coding sequence in the knockout strain but present in the wild-type control (gene check). The insertion of HXGPRT into the correct locus is verified by PCR from upstream of the gene of interest into the selectable marker (KO check) which is present only in the knockouts (PruΔ*ku80* genomic DNA used as positive control).

## GRA55 is not important for *in vitro* growth or virulence but affects chronic infection of Type II *T. gondii*

We engineered GRA55-complemented parasites by re-introducing the wild-type gene driven from its endogenous promoter into the uracil phosphoribosyltransferase (*UPRT*) locus of the *Δgra55* parasite genome [51] (Fig 4A). We verified that GRA55 targeted properly to the PV by IFA and showed similar expression levels to wild type parasites by western blot and thus we named this strain GRA55c (Fig 4B and 4C). We verified that that *Δgra55* parasites did not have any decreased growth by plaque assays (Fig 4D) compared to wild-type and complemented strains, confirming that this protein does not meaningfully participate in parasite growth *in vitro*. Upon examination of the role of GRA55 in acute infection in mice, we found that *Δgra55* parasites killed mice in a dose-dependent manner, identical to that of mice infected with wild-type and GRA55c (Fig 4E, 4F and 4G). We then conducted an investigation into

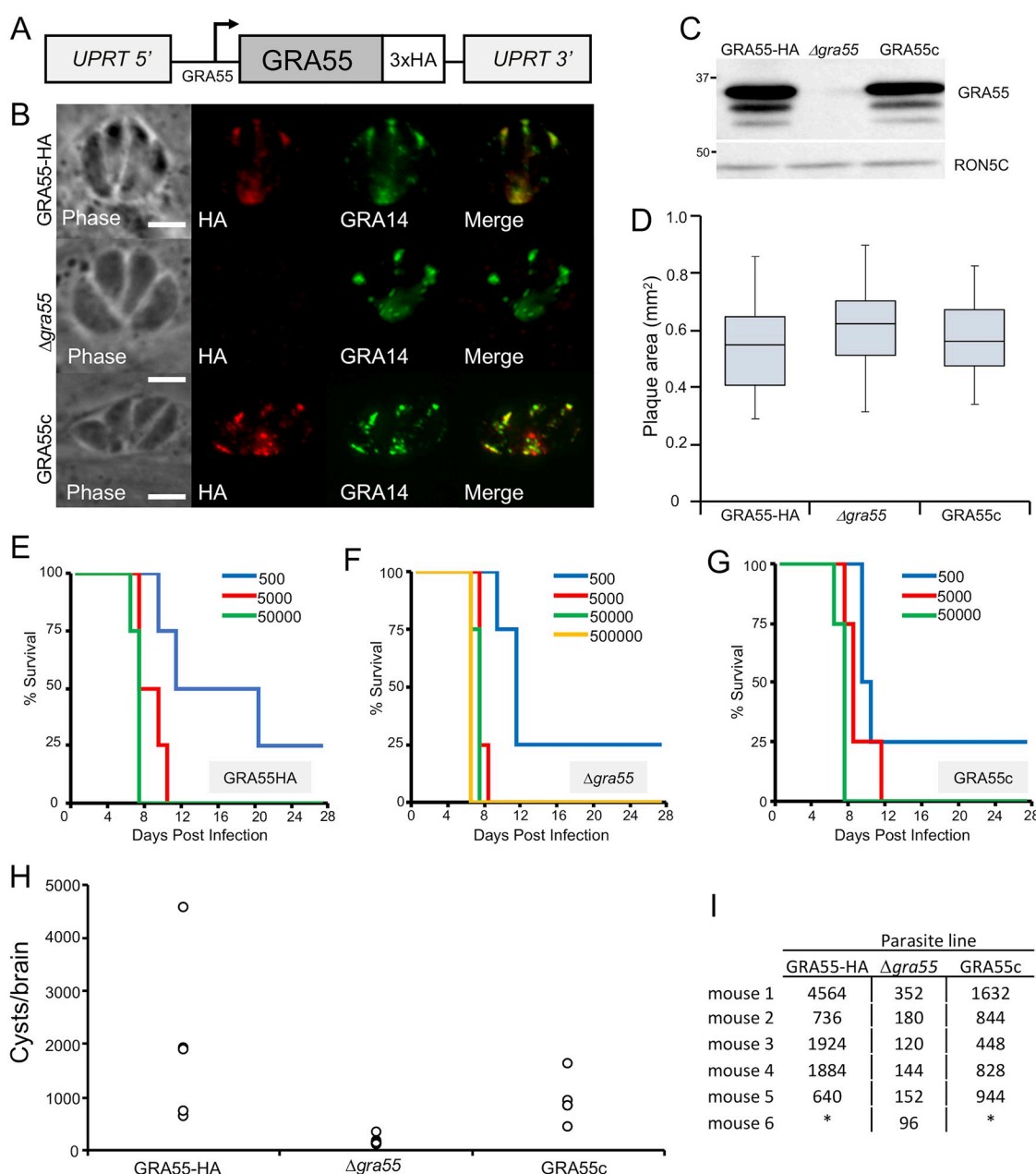

**Fig 4. GRA55 is a novel GRA which is not important for *in vitro* growth or short-term virulence.** (A) Complementation strategy of GRA55 into the UPRT locus. The complemented gene contains a C-terminal 3xHA epitope tag and is driven by the native promoter. (B) IFA of GRA55-HA, Δ*gra55*, and GRA55c parasites. Rabbit anti-HA shows strong staining in the vacuole of GRA55-HA and GRA55c parasites, colocalizing with GRA14. HA signal is absent in Δ*gra55* parasites. Scale bar: 5 μm. (C) Western blot demonstrating GRA55-HA migrating at the predicted size of 35.2 kDa (the size of GRA55-HA without the predicted signal peptide) and showing similar expression levels to the endogenously tagged strain. As expected, there is no HA signal in the Δ*gra55* strain. RON5C is used as a loading control [50]. (D) Quantitation of plaques formed by GRA55-HA, Δ*gra55*, and GRA55c parasites. Results are shown as box-whisker plot with the middle line representing the median, the bottom and top of box representing the 25th and 75th percentile, and whiskers corresponding to smallest and largest plaques. (E, F, G) C57BL/6 female mice were infected with indicated doses of GRA55HA, Δ*gra55*, and GRA55c parasites and Kaplan-Meier survival curves were generated. There was no statistical difference between the virulence of Δ*gra55* (F) vs. GRA55-HA (E) or GRA55c (G) parasites. (H) CBA/J mice were infected with 250 parasites each of the indicated strain. Mice were euthanized and mouse brains were examined by IFA after 30 days infection to enumerate *T. gondii* cyst burden. Δ*gra55* caused 10-fold lower burden of cysts when compared to GRA55-HA infection and GRA55c infection (p < 0.05 by two-tailed T-test). (I) Numerical counts of cysts per mouse brain (* = mouse dead prior to end of experiment).

whether GRA55 was important for chronic toxoplasmosis in mice. We injected 3 separate groups of mice intraperitoneally with 250 parasites per mouse, allowed the infection to proceed to 30 days, and then euthanized the mice and examined tissue cyst burden in the brains. We found that mice infected with *Δgra55* parasites suffered a consistent 10-fold decrease in cyst burden compared to the mice infected with wild-type and GRA55c parasites (Fig 4H and 4I). In summary, this data demonstrated that although GRA55 is not important for the acute infection and mortality, it is an important secreted protein for the establishment or maintenance of the chronic infection in mice.

## Discussion

Toxoplasma bradyzoites are characterized by slow growing parasites encased in a cyst wall that enables transmission of the parasite to new hosts upon predation [25,52,53]. While the cysts are formed from secreted components from the parasite, the protein composition of the cyst has been challenging to determine as it is difficult to purify robust numbers of cysts and then separate the secreted components from the parasites. In addition, while switching from tachyzoites to bradyzoites can be accomplished *in vitro*, rapidly growing tachyzoites in the culture routinely overwhelm the slow growing bradyzoites and further complicate purifications. The use of the CST1 monoclonal antibody to purify fragments of the cyst wall and identify the purified components has been successful at identifying many cyst constituents [33]. However, purifying wall fragments requires relatively low stringency lysis conditions that may result in some contamination in these purifications. The adaptation of proximity biotinylation in *T. gondii* provides a new approach to label proteins prior to isolation and is ideal for compartments hard to access like the secreted GRAs of the cyst. We were additionally able to improve the *in vitro* switch efficiency using a strain which disrupted for the bradyzoite repressor AP2IV-4 [38]. While this approach helped us identify new GRAs, we found that challenges still remain with significant tachyzoite contamination, particularly at the high multiplicity of infection which we used in these experiments to minimize host biotinylated protein contamination in our streptavidin pulldowns. The recent identification of a master regulator of bradyzoite differentiation (BFD1) is likely to enable more robust cyst production without the stress of *in vitro* stage switching culture conditions and may yield a deeper insight into the cyst composition using proximity labeling [54].

As expected, our proximity labeling experiments with MAG1 as a bait protein identified many of the same proteins we found in the tachyzoite proteome. While MAG1 is one of the earliest described bradyzoite-enriched GRAs that is abundant in cysts, similar to CST1 there is also substantial MAG1 expression in tachyzoites (both proteins ranked high in our GRA--BioID of tachyzoites) [36]. Regardless, we were able to find many of the known bradyzoite upregulated proteins including CST1, BPK1, and the newly discovered CST proteins CST2, CST3, CST4, and CST5 [30,33,34]. We also found a group of candidate GRAs, five of which we verified by endogenous gene tagging. Interestingly, each of these new GRA proteins could be clearly detected in tachyzoites. This agrees with CST2, CST3, CST 4, and CST6 recently identified in the CST1 pulldown experiments which can also be observed in tachyzoites [33]. This may be due to the *in vitro* bradyzoite switching conditions used in these experiments and could indicate that these proteins are more important for transitioning to the bradyzoite stage rather than maintenance of mature cysts. In agreement with *in vitro* cysts not containing the full complement of proteins present in *in vivo* cysts, neither this study nor the CST1 pulldown identified BCP1, which has only been identified in the cyst wall of *in vivo* derived cysts [55]. It will be interesting to determine if cysts induced via BFD1 more faithfully replicate *in vivo* cysts, and thereby enable new categories of bradyzoite GRAs to be obtained, or if these are still lacking important characteristics of cysts isolated from mice.

In agreement with the *T. gondii* genome-wide CRISPR screen, we were able to disrupt all five of the new GRAs that we identified here [56]. While gene knockouts using CRISPR/Cas9 are generally efficient in Δ*ku80* strains of *Toxoplasma*, we have found that replacing the *Toxoplasma* promoter driving HXGPRT with the *Neospora* GRA7 promoter improves the frequency of knockouts, frequently reaching 100% of the transfected population.

Most GRA proteins lack any identifiable homology to known eukaryotic proteins. GRA56, which we identified in this study, is a large protein with a domain that has homology to the melibiase family proteins. Melibiase proteins are alpha-galactosidases first discovered in plants, then found in several bacteria, animal tissues and fungi [57–65]. The enzyme acts to break apart melibiose, a disaccharide composed of galactose and glucose by a 1,6-alpha bond. In many fungi including *Saccharomyces* species, this enzyme is a large glycoprotein which contains an N-terminal signal sequence and is secreted outside of the cell, similar to GRA56 identified here [66–68]. We hypothesize that *T. gondii* may use this secreted enzyme to catabolize melibiose for energy utilization after uptake into the PV. While GRA56 does have a negative (-1.13) fitness score in the genome-wide CRISPR screen [56], it is possible that this knockout has compensated by upregulating other pathways as has previously been observed in *Toxoplasma* [69].

Our pilot *in vivo* assays suggested that disruption of GRA55 resulted in a defect in cyst counts, thus we complemented the strain with the wild-type gene. We found that the knockout showed no defect in plaque size or in acute virulence *in vivo*, but the chronic infection resulted in a ~10 fold decrease in cyst number *in vivo*, which was rescued by complementation. We did not observe any gross change is cyst size or morphology, and did not observe fragility of the cysts while manipulating the samples for counting as has been seen in the CST1 deletion mutants [31]. We performed an *in vitro* examination of switching efficiency between GRA55-HA, Δ*gra55*, and GRA55c parasites. Although there was a general trend of increased efficiency from GRA55-HA to Δ*gra55* parasites, we found that there was no significantly increased switch efficiency between the three strains. Further, complementation of GRA55 did not reverse the trend (S3 Fig). This suggests that the defect may be in tissue tropism to the brain or in the formation or maintenance of cysts *in vivo*, although we cannot exclude the possibility that subtle changes in growth *in vivo* over the 30 day time period used for cyst development play a role in the decreased cyst count in GRA55 knockouts. Since GRA55 lacks homology to known proteins, the mechanism of action will be best be determined by mutagenesis experiments to identify functional domains and additional proximity biotinylation experiments to identify binding partners and regulators. Together, this research improves our understanding of the composition and function of GRA proteins in bradyzoites which will ultimately lead to new therapies to treat this poorly understood yet critical stage of infection.

## Materials and methods

### Host cell and parasite cultures

PruΔ*ku80*Δ*hxgprt* (Type II, strain Prugniaud) and modified strains of *T. gondii* were used to infect human foreskin fibroblast (HFF) cells. HFFs were grown in Dulbecco's Modified Eagle's Medium supplemented with 10% fetal bovine serum and 2 mM glutamine, and maintained as previously described [70].

### Antibodies

Primary antibodies used in immunofluorescence assays (IFAs) and Western blot: mouse polyclonal anti-GRA14 (1:1000), rabbit anti-HA (Covance, 1:2000), rabbit polyclonal anti-ROP13 (1:500).

### IFA and Western blot

For IFA, HFFs were grown to confluence on coverslips and infected with *T. gondii* parasites. After 18 to 36 h, the coverslips were fixed and processed for indirect immunofluorescence as previously described [71]. Primary antibodies were detected by species-specific secondary antibodies conjugated to Alexa 594/488. The coverslips were mounted in ProLong Gold antifade reagent (ThermoFisher) and viewed with an Axio Imager Z1 fluorescence microscope (Zeiss) as previously described [41]. For Western blotting, parasites were lysed in Laemmli sample buffer (50 mM Tris-HCl [pH 6.8], 10% glycerol, 2% SDS, 1% 2-mercaptoethanol, 0.1% bromophenol blue), and lysates were resolved by SDS-PAGE and transferred onto nitrocellulose membranes. Blots were probed with the indicated primary antibodies, followed by secondary antibodies conjugated to horseradish peroxidase (HRP). Target proteins were visualized by chemiluminescence.

### Generation of Δ*ap2IV-4* parasites using CRISPR/Cas9 and homologous recombination

An sgRNA sequence in the coding sequence of *AP2IV-4* was chosen using the EuPaGDT tool (http://grna.ctegd.uga.edu/) and the gRNA scaffold forward and reverse sequences were engineered with *BsaI* ends (primers p1-p2 –S2 Table). This sequence was cloned into the pu6-Universal [48,49] plasmid at the same *BsaI* site. Separately, an HXGPRT cassette (driven by a *Neospora* GRA7 promoter) was amplified from the pJET1.2:NcGRA7pro:HXGPRT. The primers contained 39 bp homology to the 5' UTR (forward) and 3' UTR (reverse) regions of *AP2IV-4* (primers: p3-p4 –S2 Table). Prugniaud∆*ku80* parasites were transfected with 30 μg of closed-circular pU6-universal/sgRNA with 60 μg of the HXGPRT/*AP2IV-4-flanks* double-stranded oligonucleotide. The transfected parasites were grown in medium containing 50 μg /ml MPA (mycophenolic acid) and 50 μg /ml xanthine and cloned by limiting dilution. Knockout clones were verified by PCR (primers: gene check: p44-p45, KO check: p43, p51 –S2 Table).

### Generation of MAG1-BirA* fusion

To generate the MAG1-BirA* fusion, the *MAG1* gene and promoter was PCR amplified from genomic DNA (primers: p5-p6 –S2 Table). This was cloned into the p.LIC.BirA*.3xHA.DHFR vector by ligation independent cloning as previously described [35,72]. 30 μg of the construct was linearized with *Hpa*I and transfected into *PruΔku80Δhxgprt::Δap2IV-4.hxgrprt* parasites. The parasites were selected with medium containing 1 μM pyrimethamine, cloned by limiting dilution, and screened by IFA and Western blot against the HA tag. A clone expressing the fusion protein was selected and designated MAG1-BirA*:*Δap2IV-4.hxgrprt*.

### Affinity capture of biotinylated proteins

HFF monolayers were infected with parasites expressing MAG1-BirA* fusions or the parental line (*PruΔku80Δhxgprt:::Δap2IV-4.hxgrprt*) for 3h. Media was changed to bradyzoite "switch" media (DMEM with 1%FBS, 50mM HEPES, 1%PSG, pH 8.1). [35,72]. Parasites were incubated at 37° C in ambient air (0.05% CO2) for 5 days to induce switch to tissue cysts, exchanging media every 2 days to maintain alkaline pH [33,43]. 150 μM biotin was supplemented in the media on the final 24 hrs. Cyst formation was verified by expression of LDH2-GFP. Intracellular parasites were collected, washed in PBS, and lysed in radioimmunoprecipitation assay (RIPA) buffer (50mM Tris pH 7.5, 150 mM NaCl, 0.1% SDS, 0.5% sodium deoxycholate, 1% NP-40) supplemented with Complete Protease Inhibitor Cocktail (Roche) for 30 min on ice. Lysates were centrifuged for 15 min at 14,000 × *g* to pellet insoluble material and the

supernatant was incubated with Streptavidin Plus UltraLink Resin (Pierce) at room temperature for 4 h under gentle agitation. Beads were collected and washed five times in RIPA buffer, followed by three washes in 8M urea buffer (50 mM Tris-HCl pH 7.4, 150 mM NaCl). Ten percent of each sample was boiled in Laemmli sample buffer, and eluted proteins were analyzed by Western blotting by streptavidin-HRP while the remainder was used for mass spectrometry.

## Mass spectrometry

The proteins bound to streptavidin beads were reduced and alkylated via sequential 20 minute incubations of 5mM TCEP 10mM iodoacetamide at room temperature in the dark while being mixed at 1200 rpm an Eppendorf thermomixer. Proteins were then digested by the addition of 0.1μg Lys-C (FUJIFILM Wako Pure Chemical Corporation, 125–05061) and 0.8μg Trypsin (Thermo Scientific, 90057) while shaking 37˚C overnight. The digestions were quenched via addition of formic acid to a final concentration of 5% by volume. Each sample was desalted via C18 tips (Thermo Scientific, 87784) and then resuspended in 15μL of 5% formic acid before analysis by LC-MS/MS. Peptide samples were then separated on a 75μM ID, 25cm C18 column packed with 1.9μM C18 particles (Dr. Maisch GmbH) using on a 140-minute gradient of increasing acetonitrile and eluted directly into a Thermo Orbitrap Fusion Lumos instrument where MS/MS spectra were acquired by Data Dependent Acquisition (DDA). Data analysis was performed using the ProLuCID and DTASelect2 algorithms as implemented in the Integrated Proteomics Pipeline—IP2 (Integrated Proteomics Applications, Inc., San Diego, CA). Protein and peptide identifications were filtered using DTASelect and required a minimum of two unique peptides per protein and a peptide-level false positive rate of less than 1% as estimated by a decoy database strategy.

## Generation of GRA-HA parasites (epitope tagging of BioID hits)

For GRA55, GRA56 and 59, LIC methodology was used: To epitope tag the novel GRA candidates, ~1500 bp of the 3' genomic region of each gene was amplified using primers p7-p8 (GRA55), p15-16 (GRA66), p37-p38 (GRA59) (S2 Table) from PruΔ*ku80Δhxgprt* genomic DNA. The products were cloned into the p.LIC.3xHA.DHFR vector to add a 3xHA tag on the target gene as previously described [73–75]. The constructs were linearized and 50 μg of DNA was transfected by electroporation into PruΔk*u80Δhxgprt* strain *T. gondii*.

For GRA57 and GRA58, CRISPR/Cas9 approach was used: An sgRNA sequence in the coding sequence of the intended GRA protein was chosen using the EuPaGDT tool (http://grna. ctegd.uga.edu/) and the gRNA scaffold forward and reverse sequences were engineered with *BsaI* ends. This sequence was cloned into the pU6-universal plasmid at the same *BsaI* site: p21-22 (GRA57), p29-p30 (GRA58) (S2 Table). Separately, a 3xHA:*DHFR* region was amplified from the p.LIC.3xHA.DHFR plasmid. The primers contained a 40 bp homology to a region just upstream of the gene-of-interest STOP codon (forward) and a 38 bp homology to a region in the gene-of-interest 3' UTR (reverse) regions of the GRA gene: p23-24 (GRA57), p31-p32 (GRA58) (S2 Table). PrugniaudΔ*ku80Δhxgprt* parasites were transfected with 30 μg of closed-circular pU6-universal/sgRNA with 30 μg of the 3xHA-DHFR/*GOI-C-term-flanks* double-stranded oligonucleotide [49,72].

The transfected parasites were grown in media containing 1 μM pyrimethamine and selected parasites were cloned by limiting dilution. Clones were designated as GRA(X)-HA (e.g. GRA55-HA). All confirmed GRAs were examined *in silico* by BLAST to search for other *T. gondii* proteins with sequence similarity.

## Deletion of GRAs 55–59 using CRISPR/Cas9

An sgRNA sequence in the coding sequence of the intended GRA protein was chosen using the EuPaGDT tool (http://grna.ctegd.uga.edu/) and the gRNA scaffold forward and reverse sequences were engineered with *BsaI* ends. This sequence was cloned into the pU6-universal plasmid at the same *BsaI* site [48,49]. Oligos: p9-p10 (GRA55), p17-18 (GRA46), p25-26 (GRA57), p33-p34 (GRA58), p39-p40 (GRA59) (S2 Table). Separately, an HXGPRT cassette (using the *Neospora* GRA7 promoter) was amplified from the pJET1.2:NcGRA7pro:HXGPRT. The primers contained 39 bp homology to the 5' UTR (forward) and 3' UTR (reverse) regions of the GRA gene. Primers: p11-p12 (GRA55), p19-20 (GRA56), p27-28 (GRA57), p35-p36 (GRA58), p41-p42 (GRA59) (S2 Table) Prugniaud*Δku80Δhxgprt* GRA-HA-tagged parasites were transfected with 30 μg of closed-circular pU6-universal/sgRNA with 30 μg of the HXGPRT/*GRA-flanks* double-stranded oligonucleotide. The transfected parasites were grown in medium containing 50 μg /ml MPA (mycophenolic acid) and 50 μg /ml xanthine and cloned by limiting dilution. Knockout clones were selected by loss of HA tag (IFA) and verified by PCR to examine for the absence of the gene of interest and presence of the *HXGPRT* cassette at the predicted recombination site (KO check primers: p46-p50, each paired with p51), (gene check: p7-p8 [GRA55], p15-16 [GRA56], p52-p53 [GRA57], p54-55 [GRA58] p37-p38 [GRA59]) (S2 Table).

## Complementation of GRA55

GRA55-HA was re-introduced into the genome of the Δ*gra55* line in the uracil phosphoribosyltransferase (UPRT) locus [51]. This was done by amplifying the GRA55-3xHA gene as well as its native promoter from genomic DNA of the GRA55-HA parasite line (p13-p14) (S2 Table). The PCR product was cloned into the *UPRT* knockout vector with a tubulin promoter added using *PacI* and *SpeI*, forming a construct with the GRA55-3xHA cassette inserted between the *UPRT* 5' and 3' flanking regions. This plasmid was linearized using *SacI-HF* and transfected into Δ*gra55* parasites and selected with 5-Fluoro-5'-deoxyuridine (FUDR) to isolate parasites with the construct integrated at the *UPRT* locus [76]. Clones were selected by limiting dilution, examined by IFA (anti-HA antibody), and a positive clone was designated GRA55c.

## Plaque assay

Serial dilutions of parasites were infected into separate wells of a 6-well plate with an HFF monolayer and allowed to form plaques. The dilutions were calibrated to infect 100 parasites/well, 200 parasites/well and 300 parasites/well. 6 days after infection, HFF monolayers were fixed in 3.7% formaldehyde for 15 minutes, washed with PBS, dried and stained with safranin. Plaques were then visualized with a Zeiss upright light microscope (Zeiss Axio Imager Z1). Plaque areas were measured using ZEN software (Zeiss) and plaque sizes were compared. 2 separate experiments of 30 plaques for each parasite line were measured to generate a mean plaque size along with interquartile ranges. Statistical significance comparing lines was calculated using two-sample two-tailed t-test.

## Quantitation of in vitro bradyzoite differentiation efficiency in GRA55-HA, Δgra55, and GRA55c parasites

HFFs were grown on coverslips for seven days and infected with GRA55-HA, Δ*gra*55 or GRA55c parasites for 3h. Media was changed to bradyzoite "switch" media (DMEM with 1% FBS, 50mM HEPES, 1%PSG, pH 8.1). [35,72] and parasites were incubated at 37˚C in ambient air (0.05% $CO_2$) for 3 days to induce switch to tissue cysts [33,43]. The Coverslips were then

fixed for IFA analysis. To quantify bradyzoite switch efficiency, GFP positive and negative vacuoles were counted and percentages of GFP positive vacuoles were calculated. Two biological replicates were reported for each strain. Statistical significance was calculated comparing the means of GRA55-HA with Δ*gra55* or GRA55c with Δ*gra55* using a one-way ANOVA test (using $p < 0.05$ for significance).

## Mouse virulence assays

Intracellular GRA55-HA, Δ*gra*55 and GRA55c parasites mechanically liberated from infected HFF monolayers and resuspended in Opti-MEM Reduced Serum Medium (ThermoFisher) prior to intraperitoneal injection into groups of 4 female C57BL/6 mice each. Injected parasites were confirmed to be live and viable parasites by plaque assays with HFF monolayers. Mice were monitored for symptoms of infection, weight loss, and mortality for 21 days. Survival was plotted on a Kaplan-Meier curve. To confirm infection, surviving mice were sacrificed at 30 days after infection, and mouse brains were collected in aseptic fashion, homogenized, and examined by phase microscopy and fluorescence microscopy (for detection of GFP under *LDH2* promoter for parasite encystation). In parallel, 1/3 of each brain was homogenized and incubated with HFF monolayers to qualitatively assess viability of parasites.

## Mouse brain cyst quantitation

Intracellular GRA55-HA, Δgra55, and GRA55c parasites were mechanically liberated from infected HFF monolayers and resuspended in Opti-MEM prior to intraperitoneal injection into groups of 4 female CBA/J mice each. 250 parasites/mouse of GRA55-HA, Δ*gra55* and GRA55c were injected. The mice were monitored for 30 days after infection, and then sacrificed. Mouse brains were collected, homogenized and examined for *T. gondii* cysts (as above). Quantitation of cysts was performed by examining 25 μL aliquots of homogenate by fluorescence microscopy until approximately 25% (by volume) of each brain was examined. Total cyst burden was then extrapolated. Statistical significance of cyst burden between GRA55-HA, Δgra55, and GRA55c experiments was calculated by two-sample 2-tailed t-test [77].

## Animal experimentation ethics statement

Specific details of our protocol were approved by the UCLA Institutional Animal Care and Use Committee, known as the Chancellor's Animal Research Committee (protocol: 2004–005). Mice were euthanized for two purposes. In the acute infection model, they were euthanized when the animals reached a moribund state. For the chronic infection model, these mice were sacrificed at the conclusion of the experiment at 30 days postinfection. Euthanasia per AVMA guidelines—slow (20–30% per minute) displacement of chamber air with compressed $CO2$. This is followed by confirmatory method of cervical dislocation'.

## Supporting information

**S1 Raw images.**
(PDF)

**S1 Fig. Deletion of *AP2IV-4* gene using CRISPR/Cas9.** (A) Diagram of the construct used to delete *AP2IV-4*. Cas9 induced double stranded break (DSB) is represented by the scissors. The AP2IV-4 locus was replaced with an *HXGPRT* selectable marker using homologous

recombination simultaneously with a Cas9 induced DSB. (B) PCR verification of deletion of *AP2IV-4* and confirmation of homologous recombination of the *HXGPRT* cassette in place of the deleted locus.
(TIF)

**S2 Fig. Quantitation of plaques formed by HA-tagged and GRA knockout parasites.** (A) GRA56, (B) GRA57, (C) GRA58, (D) GRA59. Results are shown as box-whisker plot with the middle line representing the median, the bottom and top of box representing the 25th and 75th percentile, and whiskers corresponding to smallest and largest plaques.
(TIF)

**S3 Fig. Quantitation of in vitro bradyzoite differentiation efficiency in GRA55-HA, knockout, and complemented parasites following three days of treatment with atmospheric $CO_2$ conditions and alkaline media.** The mean ±SD was plotted for n = 2 biological replicates, where percentage of GFP positive vacuoles were calculated for each strain. No significance (p < 0.05) was reported comparing the means of GRA55-HA with *Δgra55* or GRA55c with *Δgra55* using a one-way ANOVA test.
(TIF)

**S1 Table. Full mass spectrometry analysis of the MAG1-BioID experiment.** There are two replicate experiments labeled MAG1-BioID-A and MAG1-BioID-B. Known GRAs shown in blue. Novel GRAs highlighted in Red. GRA-control = PruΔ*ku80*:Δ*ap2iv-4*. Proteins are ranked based on peptide spectrum matches (PSM). PSM is defined as the total number of theoretically generated peptide fragmentation spectra from the database matched to the experimental mass spectrometry fragmentation spectra. Columns are sortable using arrows next to each column heading.
(XLSX)

**S2 Table. Primers used in this study.**
(DOCX)

# Acknowledgments

Our thanks to Dr. Patricia Johnson, the Microbial Pathogenesis Training Grant T32, and supporting committee which provided professional and scientific mentorship to S.M.N. and A.C. T. throughout the grant period.

# Author Contributions

**Conceptualization:** Santhosh Mukund Nadipuram, Peter John Bradley.

**Data curation:** Santhosh Mukund Nadipuram, Shima Rayatpisheh, James Akira Wohlschlegel, Peter John Bradley.

**Formal analysis:** Santhosh Mukund Nadipuram, Shima Rayatpisheh, James Akira Wohlschlegel.

**Funding acquisition:** Santhosh Mukund Nadipuram, Amara Cervantes Thind, James Akira Wohlschlegel, Peter John Bradley.

**Investigation:** Santhosh Mukund Nadipuram, Amara Cervantes Thind, Shima Rayatpisheh.

**Methodology:** Santhosh Mukund Nadipuram.

**Project administration:** Peter John Bradley.

**Resources:** Shima Rayatpisheh, James Akira Wohlschlegel.

**Software:** Amara Cervantes Thind, Shima Rayatpisheh, James Akira Wohlschlegel.

**Supervision:** James Akira Wohlschlegel, Peter John Bradley.

**Validation:** Amara Cervantes Thind.

**Writing – original draft:** Santhosh Mukund Nadipuram, Peter John Bradley.

**Writing – review & editing:** Santhosh Mukund Nadipuram, Peter John Bradley.

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
