## [Decision Letter · Decision Letter 0]

24 Jan 2020

PONE-D-19-35931

Proximity Biotinylation Reveals Novel Secreted Dense Granule Proteins of *Toxoplasma gondii* Bradyzoites

PLOS ONE

Dear Prof Bradley,

Thank you for submitting your manuscript to PLOS ONE. After careful consideration, we feel that it has merit but does not fully meet PLOS ONE’s publication criteria as it currently stands. Therefore, we invite you to submit a revised version of the manuscript that addresses the points raised by the reviewers.

Your work was reviewed by two experts and both recommended acceptance after minor changes. Specially they requested clear evidence for expression of the identified proteins in bradyzoites. I hope you can address both set of comments. In your rebuttal and revised article make it easy for me to see that you've dealt with their concerns, as that way I'll be able to make a rapid editorial decision.

We would appreciate receiving your revised manuscript by Mar 09 2020 11:59PM. To enhance the reproducibility of your results, we recommend that if applicable you deposit your laboratory protocols in protocols.io, where a protocol can be assigned its own identifier (DOI) such that it can be cited independently in the future. For instructions see: http://journals.plos.org/plosone/s/submission-guidelines#loc-laboratory-protocols

We look forward to receiving your revised manuscript.

Kind regards,

Silvia N Moreno

Academic Editor

PLOS ONE

3. Thank you for include the following ethics statement on the submission details page:

'Specific details of our protocol were approved by the UCLA Institutional Animal Care and Use Committee, known as the Chancellor’s Animal Research Committee (protocol: 2004-005).

Mice were euthanized for two purposes. In the acute infection model, they were euthanized when the animals reached a moribund state. For the chronic infection

model, these mice were sacrificed at the conclusion of the experiment at 30 days postinfection.

Euthanasia per AVMA guidelines - slow (20-30% per minute) displacement of chamber air with compressed CO2. This is followed by confirmatory method of cervical dislocation'.

 Please also include this information in the ethics statement of the Methods section of your manuscript.

"Our thanks to Dr. Patricia Johnson, the Microbial Pathogenesis Training Grant T32, and supporting committee which provided funding as well as professional and scientific mentorship to S.M.N. and A.C.T. throughout the grant period. This work was supported by NIH grants AI064616 and AI125106 to P.J.B., and no. GM089778 to J.A.W., the Ruth L. Kirschstein National Research Service Award (T32) AI007323 to S.M.N and A.T."

Reviewers' comments:

Reviewer's Responses to Questions

**Comments to the Author**

1. Is the manuscript technically sound, and do the data support the conclusions?

Reviewer #1: Yes

Reviewer #2: Partly

2. Has the statistical analysis been performed appropriately and rigorously? 

Reviewer #1: Yes

Reviewer #2: Yes

3. Have the authors made all data underlying the findings in their manuscript fully available?

Reviewer #1: Yes

Reviewer #2: Yes

4. Is the manuscript presented in an intelligible fashion and written in standard English?

Reviewer #1: Yes

Reviewer #2: Yes

5. Review Comments to the Author

Reviewer #1: The manuscript “Proximity Biotinylation Reveals Novel Secreted Dense Granule Proteins of Toxoplasma gondii Bradyzoites” by Nadipuram et al. used a MAG1-BIRa fusion to identify novel dense granule proteins expressed in bradyzoites.

Quite a lot is known about GRA effectors in tachyzoites. However, what GRA effectors are important for bradyzoite biology is largely unknown. As such the identification of novel GRA effectors important for bradyzoite biology is of interest to the field.

Comments:

It was unclear if the MAG1-BirA fusion is indeed higher expressed in bradyzoites vs. tachyzoites. Judging from the data it seems that most of the hits that were identified were already previously identified as interacting with tachyzoite GRAs. Although this is not per se a weakness it was not immediately obvious which of their hits (or even which of the described novel GRAs) are induced in bradyzoites.

For the novel GRA55 it seems also this GRA is already highly expressed in tachyzoites. The authors show it forms less cysts in vivo. At least they should also show if it has any defect in cyst formation in vitro.

Reviewer #2: Nadipuram and colleagues have produced a well-written and interesting manuscript that identifies and examines novel GRA proteins purported to be enriched in bradyzoite stages of the parasite using a powerful protein biotinylation strategy. The data are, on the whole, solid, and will no doubt lead to further characterization of the roles of these proteins (most particularly of GRA55) in tissue cysts. My major criticism of the manuscript is that the authors have not established whether the identified GRA proteins are found/enriched in bradyzoites. The authors need to include data that show whether or not these proteins are actually expressed in bradyzoites or tachyzoites (e.g. include images of verified bradyzoites and tachyzoites in Figure 2). Ideally, they would also include some western blots of proteins extracted tachyzoites and bradyzoites to determine whether the identified GRA proteins are enriched in one or the other life stage.

Major comments

Table S1 and description of the protein ranking. This needs a better description. What do the numbers in Table S1 mean? How does this correlate to the ranking system the authors mention?

Line 221 and Figure 2. “We found that 5 of these trafficked to the PV in both tachyzoites and bradyzoites (Fig. 2).” Are the images in Figure 2 of tachyzoites or of bradyzoites? This should be specified in the figure legend. To claim that the proteins are expressed in both, they also need to included images depicting the other life stage. Ideally, the authors would undertake western blotting of proteins extracted from tachyzoites vs bradyzoites and determine whether these proteins are indeed enriched in bradyzoites.

Minor

Line 51 and elsewhere. Toxoplasma (no italics) should probably read T. gondii (italics).

Line 71. Perhaps: “… and the failure of GRA4 and GRA6 to accumulate at the cyst periphery”

Line 104. Perhaps: “In this manuscript, we build on …”

Figure 1 general comment. To verify correct insertion of the BirA* tag into the MAG1 locus, the authors should include a western blot depicting the mass of this fusion protein.

Figure 1B and Lines 134-139. Does this figure show tachyzoites (as implied in Line 134) or bradyzoites (as implied in Line 139)? The authors should clarify this.

Line 137. What do the authors mean by “the fusion expressed more robustly”? The meaning here is unclear.

Lines 145 and 148 (comment applies to other figures in the manuscript as well). Check that the scale bar is correct. If the bar really is 10 µm, the parasites would be ~20 µm long, which seems unlikely.

Line 236-238. “The other 4 proteins ….” The authors could consider including these localizations as supplementary data.

6. PLOS authors have the option to publish the peer review history of their article (what does this mean?). If published, this will include your full peer review and any attached files.

Reviewer #1: No

Reviewer #2: No

---

## [Author Response · Author response to Decision Letter 0]

15 Apr 2020

Changes made by authors:

1. New GRAs which were published in the interim (MAG2, GRA54) added to tables 1 and S1 

2. Figure S1 corrected: PCR results (stained agarose gel): inadvertently inverted – new version corrected (raw image provided as well)

--Done as requested: Title page as well as body: all style requirements changed

2. PLOS ONE now requires that authors provide the original uncropped and unadjusted images underlying all blot or gel results reported in a submission’s figures or Supporting Information files. 

--The uncropped images have been added as requested. Sent as PDF: S4_raw_images

Blot/gel images are supplied in the Supporting Information section

3. Thank you for include the following ethics statement on the submission details page:

'Specific details of our protocol were approved by the UCLA Institutional Animal Care and Use Committee, known as the Chancellor’s Animal Research Committee (protocol: 2004-005).

Mice were euthanized for two purposes. In the acute infection model, they were euthanized when the animals reached a moribund state. For the chronic infection

model, these mice were sacrificed at the conclusion of the experiment at 30 days postinfection.

Euthanasia per AVMA guidelines - slow (20-30% per minute) displacement of chamber air with compressed CO2. This is followed by confirmatory method of cervical dislocation'.

 Please also include this information in the ethics statement of the Methods section of your manuscript.

--Ethics statement placed at the end of Methods section

"Our thanks to Dr. Patricia Johnson, the Microbial Pathogenesis Training Grant T32, and supporting committee which provided funding as well as professional and scientific mentorship to S.M.N. and A.C.T. throughout the grant period. This work was supported by NIH grants AI064616 and AI125106 to P.J.B., and no. GM089778 to J.A.W., the Ruth L. Kirschstein National Research Service Award (T32) AI007323 to S.M.N and A.T."

--Please add to the funding statement: This work was supported by NIH grants AI064616 and AI125106 to P.J.B., and no. GM089778 to J.A.W., the Ruth L. Kirschstein National Research Service Award (T32) AI007323 to S.M.N and A.T. The funders had no role in study design, data collection and analysis, decision to publish, or preparation of the manuscript. 

The information provided in this statement is now the same as the funding information. The T32 grant (AI007323) is also called the Ruth L. Kirschstein Award – however, I could not fit this title into the Funding Information.

--Completed:

Data has been added for plaque assays for all novel GRAs (see new Fig. S2).

In reference to switching efficiency of AP2IV-4-KO parasites vs parental strain: we have removed the “not shown” and acknowledged previous referenced data from others. 

Reviewers' comments:

Reviewer's Responses to Questions

Comments to the Author

1. Is the manuscript technically sound, and do the data support the conclusions?

Reviewer #1: Yes

Reviewer #2: Partly 

2. Has the statistical analysis been performed appropriately and rigorously? 

Reviewer #1: Yes

Reviewer #2: Yes 

3. Have the authors made all data underlying the findings in their manuscript fully available?

Reviewer #1: Yes

Reviewer #2: Yes

4. Is the manuscript presented in an intelligible fashion and written in standard English?

Reviewer #1: Yes

Reviewer #2: Yes 

5. Review Comments to the Author

Reviewer #1: The manuscript “Proximity Biotinylation Reveals Novel Secreted Dense Granule Proteins of Toxoplasma gondii Bradyzoites” by Nadipuram et al. used a MAG1-BIRa fusion to identify novel dense granule proteins expressed in bradyzoites.

Quite a lot is known about GRA effectors in tachyzoites. However, what GRA effectors are important for bradyzoite biology is largely unknown. As such the identification of novel GRA effectors important for bradyzoite biology is of interest to the field.

Comments:

It was unclear if the MAG1-BirA fusion is indeed higher expressed in bradyzoites vs. tachyzoites. Judging from the data it seems that most of the hits that were identified were already previously identified as interacting with tachyzoite GRAs. Although this is not per se a weakness it was not immediately obvious which of their hits (or even which of the described novel GRAs) are induced in bradyzoites.

--This is an excellent point. Our choice of MAG1 as a bait protein was based on previous work indicating this protein is upregulated in bradyzoites. We acknowledge in the manuscript that there is a substantial amount of the protein in tachyzoites as well as it was highly ranked in our BioID of the tachyzoite PV.

--For the novel GRA55 it seems also this GRA is already highly expressed in tachyzoites. The authors show it forms less cysts in vivo. At least they should also show if it has any defect in cyst formation in vitro.

For GRA55, we have completed the in vitro cyst formation experiments and do not see a difference in switch efficiency. This new experimental data has been included as Fig S3. We also discuss that this difference may be due to the in vivo response to the parasites which is not possible to examine in vitro.

Reviewer #2: Nadipuram and colleagues have produced a well-written and interesting manuscript that identifies and examines novel GRA proteins purported to be enriched in bradyzoite stages of the parasite using a powerful protein biotinylation strategy. The data are, on the whole, solid, and will no doubt lead to further characterization of the roles of these proteins (most particularly of GRA55) in tissue cysts. My major criticism of the manuscript is that the authors have not established whether the identified GRA proteins are found/enriched in bradyzoites. The authors need to include data that show whether or not these proteins are actually expressed in bradyzoites or tachyzoites (e.g. include images of verified bradyzoites and tachyzoites in Figure 2). Ideally, they would also include some western blots of proteins extracted tachyzoites and bradyzoites to determine whether the identified GRA proteins are enriched in one or the other life stage.

--We agree with the reviewer and now have added images showing each of the novel GRAs present in bradyzoites (Fig 2B). We have found it very difficult to compare tachyzoite and bradyzoite levels as the secreted GRAs are confined in the PV at different levels (and switch efficiencies are still difficult), making truly comparative western analyses hard to interpret.

Major comments

Table S1 and description of the protein ranking. This needs a better description. What do the numbers in Table S1 mean? How does this correlate to the ranking system the authors mention?

--Figure legend updated to explain protein ranking: Proteins are ranked by Peptide Spectrum Matches. 

Line 221 and Figure 2. “We found that 5 of these trafficked to the PV in both tachyzoites and bradyzoites (Fig. 2).” Are the images in Figure 2 of tachyzoites or of bradyzoites? This should be specified in the figure legend. To claim that the proteins are expressed in both, they also need to included images depicting the other life stage. Ideally, the authors would undertake western blotting of proteins extracted from tachyzoites vs bradyzoites and determine whether these proteins are indeed enriched in bradyzoites.

--See comment above from reviewer 1. Now both tachyzoite and bradyzoite vacuoles are shown. We also note the challenges with interpreting the western blots of these experiments.

Minor

Line 51 and elsewhere. Toxoplasma (no italics) should probably read T. gondii (italics).

--Done as requested: All relevant references to Toxoplasma changed to T. gondii

Line 71. Perhaps: “… and the failure of GRA4 and GRA6 to accumulate at the cyst periphery”

--Changed made

Line 104. Perhaps: “In this manuscript, we build on …”

--Changed made

Figure 1 general comment. To verify correct insertion of the BirA* tag into the MAG1 locus, the authors should include a western blot depicting the mass of this fusion protein.

--Western Blot added: MAG1-BirA* fusion migrates at ~105 kDa demonstrated by anti-HA antibody probe (Fig 1E)

Figure 1B and Lines 134-139. Does this figure show tachyzoites (as implied in Line 134) or bradyzoites (as implied in Line 139)? The authors should clarify this.

--Clarified as suggested: 1B shows tachyzoite vacuoles, 1C shows bradyzoite (cyst) vacuoles.

Line 137. What do the authors mean by “the fusion expressed more robustly”? The meaning here is unclear.

--We have removed this claim

Lines 145 and 148 (comment applies to other figures in the manuscript as well). Check that the scale bar is correct. If the bar really is 10 µm, the parasites would be ~20 µm long, which seems unlikely.

--Error corrected: Scale bars updated to 5 µm and all bars resized to scale appropriately with the respective panels. 

Line 236-238. “The other 4 proteins ….” The authors could consider including these localizations as supplementary data.

--Unfortunately with recent developments, we were unable to complete these immunofluorescent stains; we have removed the gene numbers from the manuscript and the associated text mentioning their localization.

6. PLOS authors have the option to publish the peer review history of their article (what does this mean?). If published, this will include your full peer review and any attached files.

Do you want your identity to be public for this peer review? For information about this choice, including consent withdrawal, please see our Privacy Policy.

Reviewer #1: No

Reviewer #2: No

---

## [Editor Report · Decision Letter 1]

17 Apr 2020

Proximity Biotinylation Reveals Novel Secreted Dense Granule Proteins of *Toxoplasma gondii* Bradyzoites

PONE-D-19-35931R1

Dear Dr. Bradley,

We are pleased to inform you that your manuscript has been judged scientifically suitable for publication and will be formally accepted for publication once it complies with all outstanding technical requirements.

With kind regards,

Silvia N Moreno

Academic Editor

PLOS ONE

Additional Editor Comments (optional):

Thank you for revising your manuscript and for addressing the reviewers's concerns.
---

## [Editor Report · Acceptance letter]

23 Apr 2020

PONE-D-19-35931R1 

Proximity Biotinylation Reveals Novel Secreted Dense Granule Proteins of *Toxoplasma gondii* Bradyzoites 

Dear Dr. Bradley:

I am pleased to inform you that your manuscript has been deemed suitable for publication in PLOS ONE. Congratulations! Your manuscript is now with our production department. 

With kind regards,

on behalf of

Dr. Silvia N Moreno 

Academic Editor

PLOS ONE